# From Nature to Synthetic Compounds: Novel 1(N),2,3 Trisubstituted-5-oxopyrrolidines Targeting Multiple Myeloma Cells

**DOI:** 10.3390/ijms232113061

**Published:** 2022-10-27

**Authors:** Roberta Listro, Alessio Malacrida, Francesca Alessandra Ambrosio, Giacomo Rossino, Marcello Di Giacomo, Valeria Cavalloro, Martina Garbagnoli, Pasquale Linciano, Daniela Rossi, Guido Cavaletti, Giosuè Costa, Stefano Alcaro, Mariarosaria Miloso, Simona Collina

**Affiliations:** 1Department of Drug Sciences, University of Pavia, Via Taramelli 12, 27100 Pavia, Italy; 2School of Medicine and Surgery, Milan Center for Neuroscience, University of Milan Bicocca, Via Cadore 48, 20900 Monza, Italy; 3Department of Experimental and Clinical Medicine, University “Magna Græcia” of Catanzaro, Campus “S. Venuta”, Viale Europa, 88100 Catanzaro, Italy; 4Department of Earth and Environmental Sciences, University of Pavia, Via Sant’Epifanio 14, 27100 Pavia, Italy; 5Department of Health Sciences, University “Magna Græcia” of Catanzaro, Campus “S. Venuta”, Viale Europa, 88100 Catanzaro, Italy; 6Net4Science Academic Spin-Off, University “Magna Græcia” of Catanzaro, Campus “S. Venuta”, Viale Europa, 88100 Catanzaro, Italy

**Keywords:** drug resistance, drug discovery, multiple myeloma, proteasome, lactam, chiral separation

## Abstract

The insurgence of drug resistance in treating Multiple Myeloma (MM) still represents a major hamper in finding effective treatments, although over the past decades new classes of drugs, such as proteasome inhibitors and immunomodulatory drugs, have been discovered. Recently, our research team, within a Nature-Aided Drug Discovery project, isolated from *Hibiscus Sabdariffa* L. calyces the secondary metabolite called Hib-ester which possesses antiproliferative properties against human multiple myeloma RPMI 8226 cells, reduces migration and cell invasion and inhibits proteasome without neurotoxic effects. In the present study, we explored the chemical spaces of the hit compound Hib-ester. We explored the structure-activity relationships (SAR), and we optimized the scaffold through sequentially modifying Hib-ester subunits. Compound screening was performed based on cytotoxicity against the RPMI 8226 cells to assess the potential efficacy toward human MM. The ability of the most effective molecules to inhibit the proteasome was evaluated and the binding mode of the most promising compounds in the proteasome chymotrypsin binding pocket was deciphered through molecular modeling simulations. Compounds 13 and 14 are more potent than Hib-ester, demonstrating that our strategy was suitable for the identification of a novel chemotype for developing possible drug candidates and hopefully widening the drug armamentarium against MM.

## 1. Introduction

Multiple Myeloma (MM) is a hematological disorder characterized by an abnormal proliferation of clone plasma cells in bone marrow [1]. It has a great impact on society in terms of patient quality of life, and it is still considered an incurable disease due to its difficult diagnosis and its incidence and mortality [2]. Radiotherapy is just a supporting therapeutic strategy, and surgery is useful only to remove solitary plasmacytoma or for treating vertebral column compression, commonly related to MM [3]. Accordingly, chemotherapy remains the first-choice treatment, and researchers in both academia and pharma-companies are devoting intense efforts to leverage new technologies and scientific knowledge to successfully develop new effective therapies which will be able to change the patient’s quality and expectation of life. Over the past decade some progress has been made in treating MM and new classes of drugs, such as proteasome inhibitors and immunomodulatory drugs, have joined the traditional drug armamentarium (corticosteroids, alkylating agents and anthracyclines) [3]. Despite the increasing availability of effective treatments for newly diagnosed MM patients that provide longer disease-free periods, MM still progresses in a majority of cases. Therefore, finding drugs which are effective at treating relapsed/refractory multiple myeloma (RRMM) is still an urgent need [4,5].

As said before, proteasome inhibitors (PIs) are used for treating MM. In more detail, there are three drugs so far approved by the United States Food and Drug Administration (FDA) and European Medicines Agency (EMA): bortezomib (Velcade^®^); carfilzomib (Kyprolis^®^); and ixazomib (Ninlaro^®^). Second-generation PIs are under development [6,7,8].

The clinical success of proteasome inhibitors and the awareness that proteasome inhibition plays a crucial role for treating MM prompted our research team to spend some effort discovering novel anticancer drugs acting with a similar mechanism. Within a Nature-Aided Drug Discovery (NADD) project, we isolated the secondary metabolite called **Hib-ester** from *Hibiscus Sabdariffa* L. calyces. We proved that **Hib-ester** is endowed with antiproliferative properties on human multiple myeloma RPMI 8226 cells (IC_50_ value of 0.45 mg/mL at 24 h). Moreover, it is able to reduce migration and cell invasion and to inhibit proteasome without neurotoxic effects [9].

In the present study, we report the identification of a new chemotype as proteasome inhibitors, starting from **Hib-ester** as *hit* compound, exploring the chemical spaces, through sequentially modification of the structural motifs and therefore exploring the structure-activity relationships (SAR); thus optimizing the scaffold. We considered different structural decorations for investigating chemical spaces around the basic **Hib-ester** structure, considering both commercial availability of building blocks and synthetic feasibility. Accordingly, we expanded the lactone series and modified the structure of **Hib-ester** into a γ- or δ- lactam structure (Figure 1).

Most of the compounds have been synthetized as described below, whereas commercially available compounds have been purchased from suppliers. The potential role played by chirality in anticancer activity and proteasome inhibition was also investigated.

The screening of the compounds was performed based on cytotoxicity against the RPMI 8226 cells to assess the potential efficacy toward human MM and the ability of the most effective molecules to inhibit proteasome was also evaluated. Molecular modeling analysis was performed to evaluate the binding mode, and the key interactions between the most interesting compounds and the proteasome were established. Our strategy allowed the identification of a chemotype which was suitable for developing possible drug candidates.

## 2. Results and Discussion

### 2.1. Chemistry

Figure 1 reports the synthesis of the reference compound **Hib-ester**. It was synthetized in its racemic form, following the procedure reported by Kazuo et al. [10], properly modified to enhance the yield.

Briefly, after esterification of cis-aconitic acid (**I**) with sulfuric acid and methanol, we performed a stereoselective dihydroxylation of the unsaturated triester (**II**) with Upjohn’s reaction [11]. The oxidizing agent OsO_4_ was selected in consideration of its mechanism of action, which allows the syn-addition of hydroxyl groups, and for the possibility of being used in catalytic quantities thanks to the presence of stoichiometric amounts of NMO, necessary to regenerate the oxidant in situ. The oxidation of (**II**) provided the corresponding diol, showing the configuration of the two new stereocenters which were identical to that of the dimethyl ester of the (*2S,3R/2R,3S*) hibiscus acid trimethyl ester (**III**), which undergoes lactonization under microwave radiation (mw) in the absence of solvents, providing (*2S,3R/2R,3S*)-**Hib ester**.

The chemical identity of (*2S,3R/2R,3S*)-**Hib ester** was confirmed by comparing the ^1^H-NMR spectrum with the spectrum of **Hib-ester** isolated from *Hibiscus sabdariffa*.

Starting from the commercially available acid (*2R,3R/2S,3S*)-**1**, the corresponding racemic λ-lactone methyl ester (*2R,3R/2S,3S*)-**2** was obtained (Figure 2). The esterification was performed under Steglich conditions [12,13], consisting of the addition of EDC and DMAP to a methanolic solution of carboxylic acid for 18 h, and subsequent treatment with 1 M KHSO_4_ aqueous solution.

Figure 3 shows the synthesis of γ-lactam compounds **3**, **4** and **5** bearing a single stereocenter and a carboxylic function either on the C2 or C3 positions of the ring. Compound (*S*)-**3** was synthetized by treatment of the corresponding N-Boc-pyroglutamic acid methyl ester with trifluoroacetic acid (TFA) in DCM at 0 °C (Figure 3A). The synthesis of racemic (*S/R*)-N-butyl lactam methyl ester **5** (Figure 3B) was achieved starting from the corresponding commercially available lactam acid precursor (*S/R*)-**4** through the Steglich esterification procedure as previously described in Figure 2.

The procedure for obtaining N-methyl lactam esters (*S/R*)-**7** is reported in Figure 4. The commercially available methyl ester lactam (*S/R*)-**6** was methylated at the lactam nitrogen by reaction with iodomethane in presence of metallic sodium. The alkaline work-up conditions caused the simultaneous hydrolysis of the methyl ester to give the carboxylic acid V. V and was therefore re-esterified by reaction with thionyl chloride in methanol to give the final compound (*S/R*)-**7**.

A different synthetic strategy was exploited to obtain the racemic dicarboxylate lactam dimethyl ester (*2R,3R/2S,3S*)-**8**, a new γ-lactam compound structurally related to the **Hib-ester** structure (Figure 1, Figure 1), bearing two carboxylate functions (and two stereocenters) on the ring. The synthetic approach exploits 1,4-Michael’s addition of glycine imine anion to dimethyl fumarate followed by hydrolysis and concomitant intramolecular amidation/cyclization, as outlined in Figure 5. The imine (**VI**) was synthesized through condensation between pivalaldehyde and glycine methyl ester hydrochloride, in dichloromethane at room temperature. The addition of a stoichiometric amount of anhydrous MgSO_4_ to the reaction mixture, to trap the forming water, was essential to improve the yield of imine **VI**. The **VI** was condensed with dimethylfumarate in the presence of triethylamine as base and lithium bromide. Lithium ion coordinates with the nitrogen and the carbonyl oxygens of the azaenolate condensed adduct, thus improving its stability and consequently favoring the formation of the intermediate **VII**. The **VII** was directly treated with acetic acid in MeOH/H_2_O solution under reflux, thus obtaining lactam (*2R,3R/2S,3S*)-**8** as racemate and with high trans-diastereoselectivity, as expected for the higher thermodynamic stability of the main diastereomeric Michael’s adduct. 

Figure 6 reports the 2,3-*trans*-stereoselective synthesis of racemic (*2R,3R/2S,3S*) lactam 3-carboxylic acids derivatives **9**, **11**, **13**, **15**, **17** and their corresponding 3-carboxylic acid methyl esters **10**, **12**, **14**, **16**, **18**, **20**. All these compounds have been synthetized through Castagnoli–Cushman reaction (CCR), followed by Steglich esterification. Briefly, the appropriate aldehyde (**VII**) and amine were reacted in the presence of molecular sieves (4 Å, MS) to prepare the imine-intermediate (**VIII**) in situ. The **VIII** was purified from the unreacted amine using the carboxylic resin IRC50. The imine intermediate was then reacted with succinic or glutaric anhydride (for the preparation of the 5-oxopyrrolidine or 6-oxopiperidine derivatives, respectively) in *p*-xylene under reflux conditions for 10 h. The subsequent L/L extraction and purification via flash chromatography, and allowed to isolate to give the 5-oxo-pyrrolidine 3-carboxilic acid (*2R,3R/2S,3S*)-**9**, **11**, **13**, **15**. The 6-oxo-piperidine 3-carboxylic acid (*2R,R/2S,3S*) **17** and **19** underwent esterification following the same protocol reported for the preparation of compounds **2** and **5** (Figure 2 and Figure 3), thus obtaining the desired 3-carboxylic acid methyl esters **10**, **12**, **14**, **16**, **18** and **20**. (Figure 6) [14].

### 2.2. Compound ***13***: Enantiomeric Resolution and Absolute Configuration Assignment

The literature data indicate that high performance liquid chromatography (HPLC) using chiral stationary phases (CSPs) is a powerful and easily scalable tool, suitable to prepare enantiopure compounds. Based on our previous experience, this technique was applied to obtain enantiopure N-benzyl 5-oxo 2-phenyl pirrolidine carboxylic acids **13** in amounts suitable for absolute configuration assignment and for preliminary biological investigation [15,16,17,18,19,20].

A preliminary analytical screening, using the immobilized chiral HPLC columns Chiralpak IC (cellulose-based), and Chiralpak IA (amylose-based) was performed (see Appendix A). Elution conditions in both normal and reverse phase mode have been experimented, using respectively *n*-Hex in the presence of various percentages of polar modifiers (EtOH or IPA) or pure alcohols (IPA or EtOH). In both cases diethylamine (0.1%), and/or trifluoroacetic acid (0.3%) were added. For all analyses, a HPLC-UV system coupled with CD (circular dichroism) detector was used [18,21].

Chiralpack IC displayed better separation than IA in both normal and reverse phase eluting conditions (see Appendix A). Specifically, using the Chiralpak IC column and eluting with *n*-Hex/EtOH/TFA (50:50:0.3, *v*/*v*/*v*), a good enantioselectivity (α = 2.45; RS = 5.49; Table 1) in short retention times (t_R1_ = 5.15 min; t_R2_ = 7.96 min) was obtained (Figure 2).

HPLC separation of (*2R,3R/2S,3S*)-**13** was scaled up on a semipreparative scale. By using a semipreparative Chiralpak IC column (250 mm × 1 cm × 5 μm) and performing ten chromatographic runs (sample concentration 6 mg/mL, injection volume, 1 mL) 27.7 mg of first eluted enantiomer ([α]_D_^20^ = +62.3, c 0.5%, CHCl_3_) and 29.1 mg of the second one ([α]_D_^20^ = −59.2, c 0.5%, CHCl_3_) have been recovered, both with an enantiomeric excess of 99.9% (Figure 3). To sum up, our approach led to a quick access to enantiopure **13** in sufficient amounts for the preliminary biological investigation, in line with the criteria required at the initial stage of drug discovery.

To assign the absolute configuration to enantiopure **13**, the structurally related **21** was used as reference compound, since they have a similar chemical environment close to the stereocenters. Under the same elution conditions (Chiralpak IC column, eluting with n-Hex:IPA:TFA 50:50:0.3), the CD detector showed a negative peak for (*2R,3R*)-**21** and (−)-**13**, and a positive one for (*2S,3S*)-**21** and (+)-**13** (Figure 4). Taken together, the experimental evidence suggests that the (*2R,3R*) absolute configuration may be proposed for (−)-**13** and the (*2S,3S*) AC for (+)-**13.** The absolute configuration (AC) to (+)-**13** and (−)-**13** used the hyphenation of chiral HPLC with CD detector by comparison of both the elution and the sign of CD of **13** enantiomers with those of the structurally related compound (+)-(*2R,3R*)-**21** and (−)-(*2S,3S*)-**21**, under the same chiral chromatographic conditions. The AC of (+)-(*2R,3R*)-**21** and (−)-(*2S,3S*)-**21** was already assigned by our research team [22,23]. 

Therefore, the hyphenation of chiral HPLC with CD detector, together with the availability of an enantiomeric reference compound with defined AC, was successfully applied for the stereochemical assignment to enantiopure **13**. 

### 2.3. Biological Investigation and Computational Studies

We conceptualized **Hib-ester** structure as a basic molecular framework for design and development of novel agents against MM, acting through a proteasome inhibition mechanism. Our objective was to explore the minimal pharmacophoric features that are essential to interact with the proteasome.

As a first step, we experimented with the synthesis of **Hib-ester** (Figure 1), to evaluate its versatility for preparing novel analogues. However, the target product was obtained as the racemate in low yield. Moreover, the herein experimented synthesis is not suitable for the preparation of **Hib-ester** analogues, giving a limited number of building blocks which are already commercially available.

Therefore, we changed our approach and, applying the concept of the structural simplification to natural **Hib-ester**, we designed two -lactone analogues, lacking hydroxyl group in C3 position (compounds **1**, **2**) easily accessible by synthesis. To expand the compound series, we designed the most synthetic accessible 5-oxopyrrolidines and 6-oxo piperidine (compounds **3**–**20**), considering that they are generally more stable than lactones under physiological conditions, and therefore they present better developability features. Moreover, the presence of nitrogen atoms gives rise to a more functionalized scaffold. Therefore, derivatives having groups with different steric properties, such as methyl, isobutyl and benzyl moieties were designed. Lastly, we evaluated the possibility of simplifying the scaffold ring system, by eliminating one chiral center and therefore reducing the number of possible stereoisomers (compounds **3**–**7**) [24]. As a result, twenty **Hib-esters** analogs have been designed, and SAR considerations drawn. Structures, and absolute and relative configurations are reported in Table 1.

**Table 1 ijms-23-13061-t001:** Designed **Hib-ester** analog compounds: structural modification explored in the present work and the respective IC_50_ value per compounds at 24 h, determined by MTT assay in RPMI 8226 cells at different concentrations (10, 100 and 500 µM); n.e. * = compounds not effective at 500 µM.

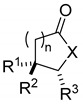 General structure of designed compounds						
**Compounds**	**X**	**n**	**R^1^**	**R^2^**	**R^3^**	**IC_50_ μM RPMI8226**
**(*2R,3R/2S,3S*)-1**	O	1	H	COOH	C_6_H_5_	n.e.*
**(*2R,3R/2S,3S*)-2**	O	1	H	COOCH_3_	C_6_H_5_	n.e.*
**(*S*)-3**	NH	1	H	H	COOCH_3_	n.e.*
**(*S/R*)-4**	N(CH_2_)_3_CH_3_	1	COOH	H	H	n.e.*
**(*S/R*)-5**	N(CH_2_)_3_CH_3_	1	COOCH_3_	H	H	n.e.*
**(*S/R*)-6**	NH	1	COOCH_3_	H	H	n.e.*
**(*S/R*)-7**	NCH_3_	1	COOCH_3_	H	H	n.e.*
**(*2R,3R/2S,3S*)-8**	NH	1	H	COOCH_3_	COOCH_3_	n.e.*
**(*2R,3R/2S,3S*)-9**	NCH_2_CH (CH_3_)_2_	1	H	COOH	C_6_H_5_	n.e.*
**(*2R,3R/2S,3S*)-10**	NCH_2_CH (CH_3_)_2_	1	H	COOCH_3_	C_6_H_5_	265 ± 49
**(*2R,3R/2S,3S*)-11**	NCH_2_CH (CH_3_)_2_	1	H	COOH	(3,5)-OCH_3_C_6_H_4_	n.e.*
**(*2R,3R/2S,3S*)-12**	NCH_2_CH (CH_3_)_2_	1	H	COOCH_3_	(3,5)-OCH_3_C_6_H_4_	263 ± 41
**(*2R,3R/2S,3S*)-13**	NCH_2_C_6_H_5_	1	H	COOH	C_6_H_5_	169 ± 21
**(*2R,3R/2S,3S*)-14**	NCH_2_C_6_H_5_	1	H	COOCH_3_	C_6_H_5_	61 ± 12
**(*2R,3R/2S,3S*)-15**	NCH_2_C_6_H_5_	1	H	COOH	(3,5)-OCH_3_C_6_H_4_	n.e.*
**(*2R,3R/2S,3S*)-16**	NCH_2_C_6_H_5_	1	H	COOCH_3_	(3,5)-OCH_3_C_6_H_4_	138 ± 27
**(*2R,3R/2S,3S*)-17**	NCH_2_CH (CH_3_)_2_	2	H	COOH	C_6_H_5_	n.e.*
**(*2R,3R/2S,3S*)-18**	NCH_2_CH (CH_3_)_2_	2	H	COOCH_3_	C_6_H_5_	301 ± 29
**(*2R,3R/2S,3S*)-19**	NCH_2_C_6_H_5_	2	H	COOH	C_6_H_5_	n.e.*
**(*2R,3R/2S,3S*)-20**	NCH_2_C_6_H_5_	2	H	COOCH_3_	C_6_H_5_	152 ± 29

The screening of **1**–**20** was firstly performed against human RPMI 8226, evaluating the cytotoxic activity in vitro (MTT assay), to assess the potential efficacy toward human MM.

RPMI 8226 cells were treated with different concentrations of all compounds (10, 100 and 500 µM) at 24 h. The concentrations have been chosen based on the results previously obtained with **Hib-ester**. MTT assay was performed to assess cell viability and calculate the IC_50_ (Table 1). For compounds not showing a significant reduction of cell viability at the concentration of 500 µM, the IC_50_ was not calculated, and they have been considered not effective. Untreated cells were used as control (CTRL).

After 24 h of treatment, trisubstituted γ-lactams **10**, **12**, **13, 14, 16** and δ-lactams **18** and **20** showed IC_50_ values <350 µM, being at least 6-fold more active that the reference compound **Hib-ester**, evidencing that the replacement of lactone with lactam ring is a structural modification which is able to improve the cytotoxic activity. Moreover, the presence of both a hydrophobic bulky substituent (isobutyl and phenyl) linked to the nitrogen and an aromatic substituent at C2 atom was essential for the activity. Conversely, results in Table 1 show that the simplification of the scaffold, by removing one chiral center (i.e., the substituent either at C2 or C3 atom, compounds **3**–**8**) is not an allowed structural modification. 

The best performing compounds are **13**, **14**, **16** and **20** with IC_50_ values ranging from 61 to 170 µM and therefore they have been further investigated, by evaluating the RPMI 8226 cell viability by MTT assays at different times (24, 48 and 72 h, Figure 5A) as well as cell viability and death by Trypan blue vital count assay (Figure 5B). Their capability to inhibit proteasome was also evaluated, by treating RPMI 8226 cells with the same concentrations used in the MTT tests. RPMI 8226 cells treated with Bortezomib (BTZ) 1, 5 and 10 nM were used as positive control for all experiments (Appendix A).

In the MTT tests, both **13** and **14** showed a time-dependent effect, with IC_50_ values of 61, 43 and 34 µM and of 169, 85 and 15 µM at 24, 48 and 72 h, respectively (Figure 5A). Conversely, racemic **16** and **20** did not show a significant increase in efficacy over time (Figure 5A). Cell viability and death by Trypan blue vital count assay evidenced that only compounds **13**, **14** induced a significant increase in cell death at 100 µM concentrations with compound **14** being the most effective one (Figure 5B).

Regarding proteasome inhibition activity, after 24 h, all tested compounds were effective in a dose-dependent manner (Appendix A). Compounds **13** and **14** were the most effective compounds; in particular, racemic **14** showed significant inhibition even at the lowest evaluated concentration (10 µM).

Biological assays evidence that **13** and the corresponding methyl ester **14** are the most effective compounds: the first reaches the lowest IC_50_ of the whole series at 72 h, whereas the latter is the most active already at 24 h and has the most prominent effect in proteasome inhibition.

Hereinbefore, all compounds, with the only exception being **Hib-ester** and compound **3**, have been evaluated as (*S/R*) or (*2R,3R/2S,3S*) racemates. To improve the SAR, we performed a preliminary investigation of the potential enantioselectivity against RPMI 8226 cells and/or in the interaction with the proteasome. In fact, studying the potential enantioselectivity in biological activity is essential to understand if novel compounds can be further developed as racemate, at least in the early stage of drug discovery, thus providing a more economic and broad ecological advantage over single enantiomers.

With an eye to the prospective development of the compound series, we performed the enantiomeric resolution of compound **13**, since its enantiomers can be easily converted into (+)-**14** and (−)-**14** by esterification under mild conditions, enabling enantioselective investigation of the second most active compound. Moreover, its enantiomers can be exploited as homochiral building blocks for the expansion of the compound series, evaluating the influence of different ester substituents.

Racemate (*2R,3R/2S,3R*)-**13** was easily resolved through preparative HPLC on chiral stationary phase and the absolute configuration (*2R,3R*) and (*2S,3S*) assigned respectively to (−)-**13** and (+)-**13** for comparison of its chiro-optical properties with an in-house available reference compound [23]. Lasty their effect on RPMI 8226 cells by MTT, Trypan blue vital count and proteasome assays have been evaluated. Obtained results are reported in Figure 6.

Enantiomer (−)-(*2R,3R*)-**13** showed a profile comparable with racemate in MTT and Trypan blue tests, whereas (+)-(*2S,3S*)-**13** showed less antiproliferative properties. Regarding proteasome inhibition, the same trend was observed, having (−)-(*2R,3R*)-**13** and (+)-(*2S,3S*)-**13** IC_50_ values of 90 ± 8 µM and 189 ± 17 µM, respectively.

Computational results highlighted that the compounds **13** and **14** are well accommodated in the proteasome chymotrypsin-like site, establishing different kinds of interactions with the key residues of the proteasome chymotrypsin pocket (Figure 7). 

By analyzing the binding mode of each enantiomer, we noticed that all compounds are involved in several interactions, such as hydrogen bond and salt bridge interactions with the proteasome chymotrypsin active site (Figure 7) and all of them bind to the pocket area in which ixazomib also binds.

In detail the (*2R,3R*)-**13** establishes hydrogen bond interactions with Lys32 and Gln131 proteasome residues (Figure 7A). Moreover, (*2R,3R*)-**13** is engaged in several hydrophobic contacts with Thr1, Arg19, Thr21, Lys32, Lys33, Ser129 and Tyr130. The (*2S,3S*)-**13** is able to establish hydrogen bond interactions with the Thr1, Lys33 and Gly47 and a salt bridge with the Lys33 (Figure 7A). In addition, the (−)-(*2S,3S*)-**13** was found well stabilized in the chymotrypsin site by means of several hydrophobic contacts with Thr1, Arg19, Thr21, Lys33, Gly47, Ala 49, Ser129, Tyr130 and Gln131.

Regarding the (*2R,3R*)-**14,** it is engaged in two hydrogen bonds, one with Ser129 and the second with Tyr130. Moreover, (*2R,3R*)-**14** is engaged in several hydrophobic contacts with Arg19, Ala20, Lys33, Ser28, Met45, Ala49, Ser53, Ser129, Tyr130 and Gln131. Considering the enantiomer (*2S,3S*)-**14**, it is able to establish a hydrogen bond with the Ala49, and a water bridge with Ala50 and additional several hydrophobic contacts with Thr1, Arg19, Ala20, Thr21, Ala27, Ala49, Ala50, Ala123, Ser129, Tyr130 and Gln131.

Moreover, the binding free energy (ΔG bind) calculations applied to assess the thermodynamic profile of the (−)-(*2R,3R*)-**13** and (+)-(*2S,3S*)-**13** and (*2R,3R*)-**14** and (*2S,3S*)-**14** enantiomers, reveal that for both of the compounds, the enantiomers (*2R,3R*)-**13** and (*2R,3R*)-**14** are associated to a stronger binding free energy with respect to (*2S,3S*)-**13** and (*2S,3S*)-**14** enantiomers (Table 2).

Considering the compound **13**, these results are in agreement with the experimental data, being (−)-(*2R,3R*)-**13** twofold more active than the opposite enantiomer in the biological assays.

Our modeling results show that both compounds **13** and **14** are able to interact with the main chains of the some pivotal residues of the proteasome chymotrypsin pocket. In addition, we notice that the (*2R,3R*)- enantiomers of both the compounds are associated to a better binding free energy with respect to the opposite enantiomers.

## 3. Materials and Methods

Reagents and solvents for synthesis, TLC and NMR were purchased from Sigma Aldrich. Silica gel for flash chromatography (60 Å, 230–400 Mesh) was purchased from Sigma Aldrich. Solvents were evaporated at reduced pressure with the Heidolph Laborota 4000 Efficient equipment. Analytical thin layer chromatography (TLC) analyses were carried out on silica gel pre-coated glass-backed plates (TLC Silica Gel 60 F254, Merk) impregnated with a fluorescent indicator, and visualized with the instrument MinUVIS, DESAGA^®^ Sastedt-GRUPPE by ultraviolet (UV) radiation from UV lamp (λ = 254 and 366 nm) or by stain reagents such as Ninidrine and Cerium Molybdate. Heating/shacking water bath (FALC Instruments, Treviglio, Italy) at 37 °C and 60 rpm. NMR were measured at room temperature (15–25 °C) on a Bruker Advance 400 MHz spectrometer, using tetramethylsilane (TMS) as internal standard and a BBI 5 mm probe. All raw FID files were processed with the Top Spin program from Bruker and the spectra were analyzed using the MestRenova 6.0.2 program from Mestrelab Research S.L. Chemical shifts are expressed in parts per million (ppm, δ scale). ^1^H-NMR spectroscopic data are reported as follows: chemical shift in ppm (multiplicity, coupling constants *J* (Hz) integration intensity). The multiplicities are abbreviated with s (singlet), d (doublet), t (triplet), q (quartet), m (multiplet) and brs (broad signal). The chemical shift of all symmetric signals is reported as the center of the resonance range. ^1^H-NMR spectroscopic data are reported as follows: chemical shift in ppm.

The purity of the final compounds were assessed by Thermo Finnigan Surveyor HPLC, with a PDA, autosampler and Surveyor pump, employing the column Phenomenex Synergi 4 µm Hydro-PR 80 Å (50 × 2.00 mm).The method applied was: Flux: 1 mL/min; Injection volume 10 µL; Wavelength of detection (λ) 254 nm; with gradient elution (solvent A: H_2_O:ACN 90:10 + 10 mM HCOONH_4_; solvent B: H_2_O:ACN 10:90 + 10 mM HCOONH_4_; gradient: 100% A in B to 100% B in 3 min, followed by isocratic elution 100% A for 4 min, return to the initial conditions in 1 min). All the final compounds had 95% or higher purity.

For the Thermo Scientific LCQ Fleet system (LCQ Fleet ion trap mass spectrometer, Surveyor MS Pump/Autosampler/PDA Detector), the method applied was: sheath gas flow rate: 8 (arb); ion spray voltage: 5 kV; capillary voltage: 37 V; capillary temperature: 275 °C; tube lens: 120 V; mass range: 100–2000 Da. Software: Thermo Scientific, Waltham, MA, USA, Xcalibur 2.2.

For the HPLC analysis and (semi)preparative separations: Jasco system (JASCO Europe, Cremella, LC, Italy) equipped with a PU-4180 plus quaternary gradient pump, 851-AS, autosampler, MD-1510 Photo Diode Array (PDA) detector and Electronic Circular Dichroism (ECD) 2095 Plus detector. Experimental data were acquired and processed by Jasco ChromNav Software. The solvent used was HPLC grade and supplied by VWR.

Optical rotation measurements were determined on a Jasco photoelectric polarimeter DIP 1000 (JASCO Europe, Cremella, LC, Italy) using a 0.5 dm cell and a sodium and mercury lamp (λ = 589 nm); sample concentration values (c) are given in % (g/100 mL).

### 3.1. Synthesis

#### 3.1.1. Dimethyl (*2S,3R/2R,3S*)-3-Hydroxy-5-oxotetrahydrofuran-2,3-dicarboxylate Dimethyl Ester (*2S,3R/2R,3S*)-Hib Ester

Firstly, *trans*-aconitic acid (58 mg, 0.35 mmol) (**I**) was subjected to magnetic stirring in methanol + 1% H_2_SO_4_ (1 mL) overnight at 60 °C under N_2_ atmosphere. The reaction mixture was concentrated in vacuo, dissolved in ethyl acetate (10 mL) and washed with water (3 × 10 mL). The organic layers were dried over Na_2_SO_4_, filtered and the solvent evaporated in vacuo. The crude trimethyl ester (**II**)was purified by flash chromatography eluting with Hex/AcOEt (7:3, *v*/*v*), giving as colorless oil (50 mg, 69%).

50 mg of (**II**) (0,23 mmol) was solubilized in 2 mL of acetone, then 13 mL of water, 41 mg on N-methyl-morpholine-N-Oxide (NMO, 0.35 mmol) and a catalytic amount of OsO_4_ (270 mL of solution 2.5% in butanol) were added. The yellow solution so obtained was kept under magnetic stirring at room temperature for 5 h. Once turned black, the solution was evaporated under reduced pressure and purified by chromatography (MP: Hex/AcOEt 4.5:5.5, *v*/*v*) to obtain 29 mg of pure diol compound (**III**) as colorless oil (29 mg, 51%).

Finally, the desired compound was obtained by adsorbing (**III**) on silica gel and subjecting it to microwave heating (2 min ramping, 5 min hold time, maximum pressure 120 psi, maximum potency 150 W, temperature 50 °C, 3 cycle). Silica was then suspended in DCM, left overnight under magnetic stirring and then filtered. The organic solution was evaporated under reduced pressure, obtaining pure (*2S,3R/2R,3S*)-**Hib Ester** (**IV**) (3.5 mg, 10%). *Rf* = 0.46 (TLC: DCM/MeOH, 90/10, *v*/*v*)

^1^H NMR (400 MHz, CDCl_3_) = δ 6.99 (s, 1H), 3.99 (s, 2H), 3.84 (s, 3H), 3.79 (s, 3H), 3.72 (s, 3H); MS-ESI *m*/*z*: calcd for C_10_H_10_O_7_ [M + H]^+^: 219.16; found: 219.21.

#### 3.1.2. Methyl (*2S,3R*)-5-Oxo-2phenyltetrahydrofuran-3-Carboxylate (*2S,3R/2R,3S*)-2

The racemic 2-pheny-lacton-3-methyl carboxylate (*2S,3R/2R,3S*)-**2** was prepared by adding 1-ethyl-3-(3-dimethylaminopropil) carbodiimide (EDC, 1.5 eq.) and 4-dimethylaminopyridine (DMAP, 0.5 eq.) to a dichlorometane solution of commercially available 2-phenyl γ-lactam 3-carboxylic acid (*2S,R/2R,3S*)-**1**, followed by the addition of MeOH (2.5 eq.) according to the general esterification procedure (Steglich esterification) described below.

The desired compound **2** was obtained as transparent oil (46.2 mg, 98%); *Rf* = 0.32 (TLC: Hex/AcOEt, 80/20, *v*/*v*).

^1^H-NMR (400 MHz, CDCl_3_) δ: 7.39–7.15 (m, 10H, Ar, mixture of diasteromers), 5.68 (d, *J* = 7.9 Hz, 1H, diasteromer A, CH), 5.59 (d, *J* = 7.1 Hz, 1H, CH, diasteromer B), 3.71 (s, 3H, CH_3_, diasteromer A), 3.70–3.62 (m, 1H, CH mixture of diasteromers), 3.32–3.23 (m, 1H, CH, mixture of diasteromers), 3.22 (s, 3H, CH_3_, diasteromer B), 3.06–2.65 (m, 4H, CH_2_, mixture of diasteromers); MS-ESI *m*/*z*: calcd for C_12_H_12_O_4_ [M + H]^+^: 221.08; found: 221.19.

#### 3.1.3. (S)-Methyl-5-oxopyrrolidine-2-carboxylate-3

The N-Boc pyroglutamic acid methyl ester (1 mmol) was solubilized in DCM (5 mL). Then, an equity part of TFA with respect to DCM (1:1) was added dropwise at 0 °C. After 15 min the reaction was dried under reduced pressure. An L/L extraction with DCM and a solution of NaHCO_3_ was performed and the washed organic phase was then dried with Na_2_SO_4_ anhydrous, filtered, and evaporated to dryness at reduced pressure. Without any other purification, the pyroglutamic acid methyl ester (*S*)-**3** was obtained (26 mg, 44%); *Rf* = 0.42 (TLC: DCM/MeOH/NH_3_ in MeOH, 90/10/0.3, *v*/*v*/*v*).

^1^H NMR (C_3_D_6_O) δ: 4.20.4.10 (m, 1H, NH), 3.65–3.55 (m, 1H, CH2), 3.20–3.15 (m, 1H, CH_2_), 2.40–2.20 (m, 2H, CH_2_), 2.00–1.90 (m, 3H, CH_3_); MS-ESI *m*/*z*: calcd for C_6_H_9_NO_3_ [M + H]^+^: 144.14; found: 144.08.

#### 3.1.4. Methyl 1-Butyl-5-oxopyrrolidine-3-carboxylate (*S/R*)-5

The γ-lactam 3-methyl carboxylate (*S,R*)-**5** was synthesized by adding 1-ethyl-3-(3-dimethylaminopropil) carbodiimide (EDC,1.5 eq.) and 4-dimethylaminopyridine (DMAP, 0.5 eq.) to a DCM solution of commercial available γ-lactam 3-carboxylic acid (*S,R*)-**4**, followed by the addition of MeOH (2.5 eq.) according to the general esterification procedure (Steglich esterification) described below.

The desired lactam **5** was obtained as transparent oil (38.4 mg, 99%); *Rf* = 0.34 (TLC: Hex/AcOEt, 50/50, *v*/*v*).

^1^H-NMR (400 MHz, CDCl_3_) δ: 3.67 (s, 3H, CH_3_), 3.58–3.46 (m, 2H, CH_2_), 3.29–3.11 (m, 3H, CH_2,_ CH), 2.68–2.54 (m, 2H, CH_2_), 1.49–1.40 (m, 2H, CH_2_), 1.32–1.20 (m, 2H, CH_2_), 0.83 (t, 2H, CH_2,_
*J* = 7.25); MS-ESI *m*/*z*: calcd for C_10_H_17_NO_3_ [M + H]^+^: 200.25; found: 200.42.

#### 3.1.5. Methyl 1-Methyl-5-oxopyrrolidine-3-carboxylate (*S/R*)-7

To a commercially available lactam 3-carboxylic acid methyl ester **6** solution in 10 mL of toluene anhydrous, at room temperature and anhydrous condition, metallic sodium (1.5 equiv.) was added. After 30 min, the methyl iodine (1 equiv.) was added and the reaction mixture was maintained under magnetic stirring overnight. Then, the quenching was performed with MeOH and after effervescence the solvent was evaporated. Due to the high solubility of the desired compound in water, no purification was performed.

The crude carboxylate product (**V**) thus obtained was reacted with 2 mL of thionyl chloride (2 equiv.) in 20 mL of MeOH and the mixture was maintained under magnetic stirring overnight. The purification was performed via a chromatographic column with DCM/MeOH 9:1 as mobile phase and the desired product was obtained with 23% of yield (20 mg); *Rf* = 0.62 (TLC: DCM/MeOH, 90/10, *v*/*v*).

^1^H-NMR (400 MHz, CDCl_3_) δ: 3.68 (s, 3H, CH_3_), 3.60–3.48 (m, 2H, CH_2_), 2.80 (s, 3H, CH_3_), 2.65–2.60 (m, 2H, CH_2_); MS-ESI *m*/*z*: calcd for C_7_H_11_NO_3_ [M + H]^+^: 158.17; found: 158.23.

#### 3.1.6. Methyl (E)-2-((2,2-Dimethylpropylidene) Amino) Acetate (E)-VI

To a solution of glycine HCl (1 equiv.) in DCM (10 mL), TEA (1.1 equiv.) and MgSO_4_ (2 equiv.) were added at room temperature, preserving the anhydrous conditions. After 1 h, 2,2 dimethyl propanal (pivalaldehyde) (1.1 equiv.) was then added and left to react for 8 h at room temperature. The reaction was filtered to remove the white solid and then solubilized and extracted with DCM and a solution of NaHCO_3_. The washed organic phase was then dried with Na_2_SO_4_ anhydrous, filtered and evaporated to dryness at reduced pressure to give **VI** as a transparent liquid, (97 mg, 61%); *Rf* = 0.52 (TLC: Hex/AcOEt, 80/20, *v*/*v*).

^1^H-NMR (400 MHz, CDCl_3_) δ: 4.05 (s, 2H, CH_2_), 3.80 (s, 3H, CH_3_), 1.00 (s, 9H, CH_3_); MS-ESI *m*/*z*: calcd for C_8_H_15_NO_2_ [M + H]^+^: 156.11; found: 156.21.

#### 3.1.7. Trimethyl (E)-1-((2,2-Dimethylpropylidene) Amino) Propane-1,2,3-tricarboxylate (E)-VII

The imine **VI** (2 eq) was solubilized in anhydrous THF dry and after 30 min TEA (1 eq) and a solution of LiBr (1 eq) in anhydrous THF dry were added dropwise under N_2_ atmosphere and subjected to magnetic stirring. After complete solubilization of the mixture, dimethyl fumarate (1 eq) was added at −15 °C. After 20 h, the reaction mixture was quenched with water. An L/L extraction with Et_2_O and H_2_O allowed us to obtain compound **VII** as a transparent oil (90 mg, 48%); *Rf* = 0.48 (TLC: Hex/AcOEt, 80/20, *v*/*v*).

^1^H-NMR (400 MHz, CDCl_3_) δ: 4.10 (m, 1H, CH), 3.75–3.50 (m, 9H, CH_3_), 3.10 (m, 1H, CH), 2.90 (m, 1H, CH), 2.20–2.210 (m, 1H, CH), 1.20 (s, 2H, CH_2_), 1.00–0.75 (m, 9H, CH_3_); MS-ESI *m*/*z*: calcd for C_14_H_23_NO_6_ [M + H]^+^: 302.15; found: 302.34.

#### 3.1.8. Dimethyl 5-Oxopyrolidine-2,3-dicarboxylate (*2R,3R/2S,3S*)-8

Compound **VI**) (1 eq) was solubilized in H_2_O/CH_3_OH 1:4 and the reaction mixture maintained under magnetic stirring for 48 h. After an L/L extraction with DCM and a solution of NaHCO_3_, the washed organic phase was then dried with anhydrous Na_2_SO_4_, filtered and evaporated to dryness under reduced pressure. Crude **8** was purified by flash chromatography and eluted with DCM/AcOEt 1:2, thus obtaining pure **8** with 33% of yield (20 mg); *Rf* = 0.24 (TLC: DCM/AcOEt, 1/2, *v*/*v*).

^1^H-NMR (400 MHz, C_3_D_6_O) δ: 6.80–6.10 (s, 1H, NH), 4.60–4.50 (m, 1H, CH), 3.75–3.10 (m, 6H, CH_3_), 3.40–3.30 (m, 1H, CH), 2.70–2.50 (m, 2H, CH_2_); MS-ESI *m*/*z*: calcd for C_8_H_11_NO_5_ [M + H]^+^: 202.06; found: 202.18.

### 3.2. General Procedure for the Preparation of 5- and 6-Oxo-piperidines

In a round-bottom flask aldehyde (1 eq) and amine (1 eq) were dissolved in toluene, 4 Å molecular sieves (MS) were added and the reaction mixture was stirred at room temperature. Afterwards, 4 h carboxylic resin IRC50 (10 meq/g) was added and the mixture kept under mechanical stirring for 20 min. Subsequently, MS and IRC50 was filtered off and the solvent evaporated under reduced pressure. The yellowish oil obtained was dissolved in p-xylene and the appropriate anhydride was (1 eq) added. The reaction mixture was stirred at 140 °C for 10 h and subsequently the yellow solution was evaporated in vacuo. Crude products were treated with Et_2_O and properly purified, as follows.

Method A. The solid obtained after addition of Et_2_O was recovered by filtration, washed with Et_2_O and dried under vacuum.

Method B. The precipitation with Et_2_O was allowed to separate secondary products. The supernatant was extracted via L/L extraction with DCM and 10% HCl. The organic phase was Na_2_SO_4_ and evaporated under reduced pressure. Compounds obtained were purified by flash chromatography, eluting with DCM/MeOH/HCOOH 95:5:0.2.

Method C. The organic phase was extracted with 10% HCl, dried with Na_2_SO_4_ and concentrated. The residue was purified by flash chromatography, eluting with DCM/MeOH/HCOOH 95:5:0.2

All compounds were obtained with suitable purity (>95%).

#### 3.2.1. 1-Isobutyl-5-oxo-2-phenylpyrrolidine 3-Carboxylic Acid (*2R,3R/2S,3S*)-9

Purification via Method B, with solid (99 mg, 40%), *Rf* = *0.37* (DCM/MeOH 95:5 with 0.2% of formic acid; *v*/*v*/*v*), mp 120.6–122.7 °C; ^1^H-NMR (400 MHz, CDCl_3_) δ: 7.25–7.50 (m, 5H, Ar), 5.0 (s, 1H, CH), 3.5–3.30 (m, 1H, CH) 3.20–3.10 (m, 1H, CH), 2.80–2.60 (m, 2H, CH_2_), 2.40–2.30 (m, 1H, CH), 1.75–1.90 (m, 1H, CH), 0.90–0.80 (m, 6H, CH_3_); MS-ESI *m*/*z*: calcd for C_15_H_21_NO_3_ [M − H]^−^: 260.14; found: 260.32.

#### 3.2.2. 2-(3,5-Dimethoxyphenyl)-1-isobutyl-5-oxopyrrolidine-3-carboxylic Acid (*2R,3R/2S,3S*)-11

Purification via Method B, yellow oil (92 mg, 24%), *Rf* = 0.35 (DCM/MeOH 95:5 with 0.2% of formic acid; *v*/*v*/*v*), ^1^H-NMR (400 MHz, CDCl_3_) δ: 6.80 (s, 3H, Ar), 4.25–4.20 (m, 1H, CH), 3.60–3.75 (m, 6H, CH_3_), 3.30–3.20 (m, 1H, CH), 2.70–2.45 (m, 2H, CH_2_) 2.30–2.25 (m, 1H, CH), 1.85–1.90 (m, 2H, CH_2_) 0.75–0.60 (m, 6H, CH_3_)O; MS-ESI *m*/*z*: calcd for C_17_H_23_NO_5_ [M − H]^−^: 320.16; found: 320.37.

#### 3.2.3. 1-Benzyl-5-oxo-2-phenylpyrrolidine 3-Carboxylic Acid (*2R,3R/2S,3S*)-13

Purification via Method A, white solid (396 mg, 28%). *Rf* = 0.34 (DCM/MeOH 95:5 with 0.2% of formic acid; *v*/*v*/*v*), mp 169.6–170.2 °C; ^1^H-NMR (400 MHz, CDCl_3_) δ: 7.45–7.00 (m, 10H, Ar), 4.80–4.90 (m, 1H, CH), 4.70–4.60 (m, 1H, CH), 3.50–3.40 (m, 2H, CH_2_), 2.90–2.60 (m, 2H, CH_2_); ^13^C NMR (100 MHz, CD_3_OD) δ: 173.00, 172.20, 139.96, 136.52, 129.95, 128.42, 140.16, 128.24, 128.01, 127.14, 63.73, 45.47, 44.86, 33.02; MS-ESI *m*/*z*: calcd for C_18_H_17_NO_3_ [M − H]^−^: 294.12; found: 294.34.

#### 3.2.4. 1-Benzyl-2-(3,5-dimethoxyphenyl)-5-oxopyrrolidine-3-carboxylic Acid (*2R,3R/2S,3S*)-15

Purification via Method C, yellow oil (102 mg, 37%), *Rf* = 0.37 (DCM/MeOH 95:5 with 0.2% of formic acid; *v*/*v*/*v*), ^1^H-NMR (400 MHz, CDCl_3_) δ: 7.50–6.20 (m, 8H, Ar), 5.00–4.80 (m, 1H, CH), 4.50 (s, 1H, CH), 3.75–3.40 (m, 6H, CH_3_), 3.00–2.60 (m, 2H, CH_2_) 1.21–0.75 (m, 2H, CH_2_); MS-ESI *m*/*z*: calcd for C_20_H_21_NO_5_ [M − H]^−^: 355.14; found: 355.39.

#### 3.2.5. 1-Isobutyl-6-oxo-2-phenylpiperidine 3-Carboxylic Acid (*2R,3R/2S,3S*)-17

Purification via Method C, with solid (89 mg, 22%), *Rf* = 0.34 (DCM/MeOH 95:5 with 0.2% of formic acid; *v*/*v*/*v*), mp 133.3–134.6 °C; ^1^H-NMR (400 MHz, CDCl_3_) δ: 7.50–7.10 (m, 6H, Ar), 5.20 (s, 1H, CH), 4.10–3.90 (m, 1H, CH), 2.90 (s, 1H, CH), 2.80–2.50 (m, 2H, CH_2_), 2.30–1.70 (m, 4H, CH_2_), 1.00–0.75 (m, 6H, CH_3_); MS-ESI *m*/*z*: calcd for C_17_H_23_NO_3_ [M − H]^−^: 288.17; found: 288.38.

#### 3.2.6. 1-Benzyl-6-oxo-2-phenylpiperidine 3-Carboxylic Acid (*2R,3R/2S,3S*)-19

Purification via Method A, with solid (73 mg, 28%), *Rf* = 0.34 (DCM/MeOH 95:5 with 0.2% of formic acid; *v*/*v*/*v*), mp 167.8–168.9 °C; ^1^H-NMR (400 MHz, CDCl_3_) δ: 7.45–7.05 (m, 10H, Ar), 5.55–5.50 (m, 1H, CH), 5.00–4.90 (m, 1H, CH) 3.6–3.3(m, 2H, CH2) 2.90–2.10 (m, 2H, CH_2_), 2.10–1.90 (m, 2H, CH_2_); MS-ESI *m*/*z*: calcd for C_20_H_21_NO_3_ [M − H]^−^: 322.15; found: 322.39.

### 3.3. Esterification Procedure

In a two-neck round-bottom flask, preserving the anhydrous conditions, a solution of starting acids (1 mmol) in dry DCM (3 mL) was prepared. 1-ethyl-3-(3-dimethylaminopropyl) carbodiimide (EDC, 1.5 eq) and 4-dimethylaminopyridine (DMAP, 0.5 eq) were added to the solution at 0–4 °C. After 5 min, methanol (2.5 eq) was added, then the mixture was left reacting at room temperature overnight. The workup procedure followed three washing episodes with KHSO_4_ 1M (3 × 10 mL). After that, the organic phase was dried with anhydrous Na_2_SO_4_, filtrated, and evaporated to dryness under reduced pressure. Thus, the pure methyl ester compounds were obtained in quantitative yield and suitable purity (>95%).

#### 3.3.1. Methyl 1-Isobutyl-5-oxo-2-phenylpyrrolidine-3-carboxylate (*2R,3R/2S,3S*)-10

Yellow oil (50.1 mg, 99%). *Rf* = 0.62 (DCM/MeOH 95:5; *v*/*v*), ^1^H-NMR (400 MHz, CDCl_3_) δ:7.35–7.10 (m, 5H, Ar), 4.85 (d, 1H, CH, *J* = 4.79), 3.66 (s, 3H, CH_3_), 3.48–3.40 (m, 1H, CH) 3–2.91 (m, 1H, CH), 2.81–2.67 (m, 2H, CH_2_), 2.35–2.28 (m, 1H, CH), 1.82–1.70 (brs, 1H, CH), 0.78 (m, 6H, CH_3_); MS-ESI *m*/*z*: calcd for C_16_H_21_NO_3_ [M + H]^+^: 276.15; found: 276.28.

#### 3.3.2. Methyl 2-(3,5-Dimethoxyphenyl)-1-isobutyl-5-oxopyrrolidine-3-carboxylate (*2R,3R/2S,3S*)-12

Transparent oil (51.2 mg, 98%). *Rf* = 0.58 (DCM/MeOH 95:5; *v*/*v*), ^1^H-NMR (400 MHz, CDCl_3_) δ: 6.38–6.20 (m, 3H, Ar), 4.78 (d, *J* = 5.2 Hz, 1H), 3.72 (s, 6H, CH_3_), 3.67 (s, 3H, CH_3_), 3.53–3.39 (m, 1H, CH_2_), 3.03–2.89 (m, 1H,), 2.83–2.63 (m, 2H, CH_2_), 2.44–2.31 (m, 1H, CH_2_), 1.85–1.68 (m, 1H, CH), 0.84–0.69 (m, 6H, CH_3_); MS-ESI *m*/*z*: calcd for C_18_H_25_NO_5_ [M + H]^+^: 336.40; found: 336.17.

#### 3.3.3. Methyl 1-Benzyl-5-oxo-2-phenylpyrrolidine-3-carboxylate (*2R,3R/2S,3S*)-14

Yellow oil (49 mg, 99%). *Rf* = 0.57 (DCM/MeOH 95:5; *v*/*v*), ^1^H-NMR (400 MHz, CDCl_3_) δ:7.35–6.90 (m, 10H, Ar), 5.05 (d, 1H, CH_2_, *J* = 14.72), 4.54 (d, 1H, CH, *J* = 5.67), 3.57 (s, 3H, CH_3_), 3.44–3.39 (d, 1H, CH_2_, *J* = 14.72), 3.03–2.96 (m, 1H, CH), 2.86–2.70 (m, 2H, CH_2_); ^13^C NMR (100 MHz, CDCl_3_) δ: 171.71, 171.63, 137.86, 134,59, 128.15, 127.61, 127.57, 127.39, 126.62, 125.92, 51.36, 44.94, 43.40, 32.60; MS-ESI *m*/*z*: calcd for C_19_H_19_NO_3_ [M + H]^+^: 310.37; found: 310.64.

#### 3.3.4. Methyl 1-Benzyl-2-(3,5-dimethoxyphenyl)-5-oxopyrrolidine-3-carboxylate (*2R,3R/2S,3S*)-16

Transparent oil (48 mg, 99%). *Rf* = 0.64 (DCM/MeOH 95:5; *v*/*v*), ^1^H-NMR (400 MHz, CDCl_3_) δ: 7.25–6.96 (m, 5H, Ar), 6.40–6.18 (m, 3H, Ar), 5.00 (d, *J* = 14.7 Hz, 1H, CH_2_), 4.48 (d, *J* = 6.4 Hz, 1H, CH), 3.69 (s, 6H, CH_3_), 3.58 (s, 3H, CH_3_), 3.55–3.47 (d, *J* = 14.7 Hz, 1H, CH_2_), 2.99 (m, 1H, CH), 2.86–2.66 (m, 2H, CH_2_; MS-ESI *m*/*z*: calcd for C_21_H_23_NO_5_ [M + H]^+^: 370.16; found: 370.42.

#### 3.3.5. Methyl 1-Isobutyl-6-oxo-2-phenylpiperidine-3-carboxylate (*2R,3R/2S,3S*)-18

Yellow oil (37 mg, 99%). *Rf* = 0.58 (DCM/MeOH 95:5; *v*/*v*), ^1^H-NMR (400 MHz, CDCl_3_) δ:7.33–7.08 (m, 5H, Ar), 5.03 (d, 1H, CH, *J* = 3.47Hz), 3.94–3.86 (m, 1H, CH_2_), 3.68 (s, 3H, CH_3_), 2.80–2.75 (m, 1H, CH), 2.63–2.53 (m, 1H, CH_2_), 2.46–2.38 (m, 1H, CH_2_), 2.12–2.05 (m, 1H, CH_2_), 2.02–1.88 (m, 2H, CH_2_), 1.84–1.76 (m, 1H, CH), 0.82–0.76 (m, 6H, CH_3_); MS-ESI *m*/*z*: calcd for C_17_H_23_NO_3_ [M + H]^+^: 290.17; found: 290.38.

#### 3.3.6. Methyl 1-Benzyl-6-oxo-2-phenylpiperidine-3-carboxylate (*2R,3R/2S,3S*)-20

White oil (36.5 mg, 98%). *Rf* = 0.59 (DCM/MeOH 95:5; *v*/*v*), ^1^H-NMR (400 MHz, CDCl_3_) δ: 7.38–7.17 (m, 6H, Ar), 7.1–7.04 (m, 4H, Ar), 5.51 (d, *J* = 12.0 Hz, 1H, CH_2_), 4.82 (d, *J* = 3.8 Hz, 1H, CH), 3.41 (s, 3H, CH_3_), 3.26 (d, *J* = 12.0 Hz, 1H, CH_2_), 2.74–2.62 (m, 2H, CH_2_), 2.57–2.45 (m, 1H), 2.00–1.79 (m, 2H, CH_2_); MS-ESI *m*/*z*: calcd for C_20_H_21_NO_3_ [M + H]^+^: 324.15; found: 324.39.

### 3.4. HPLC Chiral Resolution

For the analytical screening, Chiralpak^TM^ IC (cellulose tris-3,5-dichlorophenylcarbamate immobilized on silica gel, 0.46 cm diameter × 25 cm length, 5 µm), Chiralpak^TM^ IA (amylose tris-3,5-dimethylphenylcarbamate, immobilized on silica gel, 0.46 cm diameter × 25 cm length, 5 µm (all produced by Daicel Industries Ltd., Tokyo, Japan)) and Lux 5u Amylose-2^TM^ (0.46 cm diameter × 15 cm length, 5 µm (produced by Phenomenex, Torrance, CA, USA)) were used as CSPs. Different compositions of n-Hex-alcohol (IPA or EtOH) mixture or pure alcohol were used as mobile phase and DEA (0.1%) and TFA (0.3%) were added to the mobile phase for analysis on the immobilized Chiralpak^TM^ IA column. The (*2S,3R/2S,3R*)-**13** was dissolved in the mobile phase and analyzed at room temperature (injection volume 10µL; flow rate 1 mL/min —unless otherwise specified).

Retention factors of first and second eluted enantiomer k_a_ and k_b_, respectively, were calculated according to: (t_r_ − t_0_)/t_0_, where the dead time t_0_ was considered to be equal to the peak of the solvent front for each particular run. Resolution was calculated according to Ph. Eur. 2.2.29 [25] and enantioselectivity (α) was calculated according to: α = k_b_/k_a_.

For the (semi)-preparative runs, the Chiralpak^TM^ IC column (1 cm × 25 cm length, 5 μm) was used, eluting with n-Hex/EtOH/DEA/TFA acid (50:50:0.3, *v*/*v*/*v*) at a flow rate of 2.5 mL/min and a UV detection at 220 nm. The (*2S,3R/2S,3R*)-**13** was dissolved in the mobile phase (6 mg/mL) and the injection volume was 1 mL.

### 3.5. Biological Assays

#### 3.5.1. Cell Culture

Human multiple myeloma RPMI 8226 cells were cultured in RPMI 1640 medium supplemented with 10% fetal bovine serum, 1% L-glutamine and 1% penicillin and streptomycin (Euroclone, Italy). Human glioblastoma U87-MG cells were cultured in DMEM low glucose supplemented with 10% fetal bovine serum, 1% L-glutamine and 1% penicillin and streptomycin (Euroclone, Italy). Cells were incubated in a humidified incubator at 37 °C and 5% CO_2_. A stock solution of all compounds in DMSO (50 mM) was prepared and then directly diluted in culture medium.

#### 3.5.2. MTT Assay

Cells were seeded in 96-well plates at a density of 1 × 10^4^ cells/well and were treated after 24 h with RA, AM or SU molecules. After 24 h, a 3-(4,5-dimethylthiazol-2-yl)-2,5-diphenyltetrazolium bromide (MTT, Sigma Aldrich, United States) solution was added to each well to reach a final concentration of 0.5 mg/mL. After 4 h of incubation (4 h for RPMI 8226 cells), formazan salt was solubilized in ethanol and absorbance was measured at 570 nm in a microplate reader (BMG-Labtech, Ortenberg, Germany).

#### 3.5.3. Trypan Blue Cell Viability Assay

Cells were seeded in 6-well plates at a density of 2.5 × 10^5^ cells/well and were treated after 24 h with different concentrations of RA molecules. After 24 h of treatment, cells were collected and stained with Trypan blue vital dye (Sigma Aldrich, Burlington, VT, USA). Viable and dead cells were then counted in a hemocytometer under a light microscope.

#### 3.5.4. Proteasome Activity Assay

Cells not used in the Trypan blue assay were lysed to assess proteasome activity. Briefly, cells were resuspended in a lysis buffer (50 mM Hepes Ph 7.5, 150 mM NaCl, 10% glycerol, 1% Triton X-100, 1.5 mM MgCl_2_, 5 mM EGTA) and were mechanically lysed with a vortex. Obtained protein lysates were centrifuged at 13,500 RPM for 15 min and were quantified using the Bradford method. Some 40 µg of proteins were then loaded in black-96-well plate to perform the proteasome activity assay. In each well, 7.6 mg/mL proteasome substrate (N-Succinyl-Leu-Leu-Val-Tyr-7-Amido-4-Methylcoumarin, Sigma Aldrich, United States) and proteasome buffer (250 mM Hepes pH 7.5, 5 mM EDTA pH 8.0, 0.5% NP-40, 0.01% SDS) were added to the proteins. Fluorescence was measured after 2 h of incubation in a microplate reader (excitation 380 nm, emission 460 nm; BMG-Labtech, Ortenberg, Germany).

#### 3.5.5. Statistical Analysis

Data showed the mean ± standard deviation (SD) from at least three independent experiments. Statistical analysis was performed using GraphPad Prism 3 software. The differences between control and treated cells were evaluated using one way ANOVA analysis of variance followed by Dunnet’s multiple comparison test. Statistical significance was set at *p* < 0.05 and *p* < 0.01.

### 3.6. Computational Studies

The crystal structure of the Human 20S proteasome complex with Ixazomib, deposited in the Protein Data Bank (PDB) with the PDB code 5LF7 was used for our molecular modeling analysis [26]. The receptor structure was prepared by means of the Protein Preparation Wizard tool, using the same protocol reported in our previous work [27]. The reliability of our molecular recognition approach was evaluated by means of redocking calculations performed using the Glide Standard Protocol (SP) algorithm that was able to reproduce the experimentally determined binding mode with a Root Mean Square Deviation (RMSD) value equal to 0.651 Å [28]. The compounds were prepared by means of the LigPrep tool, hydrogens were added, salts were removed, ionization states were calculated using the Ionizer at pH 7.4 [29]. The docking studies were performed by means of the Glide v. 6.7 SP algorithm and 10 poses for ligands were generated [28]. For each compounds the best docking pose was subjected to molecular mechanism generalized Born surface area (MM-GBSA), using VSGB as the solvation model and OPLS_2005 as the force field [30,31].

## 4. Conclusions

In this report, we described the identification of novel 1(N),2,3 trisubstituted-5-oxopyrrolidines targeting multiple myeloma cells, by exploring the chemical spaces around the hit **Hib-ester**; a secondary metabolite extracted from *Hibiscus Sabdariffa* L. calyces. Compounds (*2R,3R/2S,3S*)-**13** and **14** resulted in the most active ones; the first one being twelve-fold more active, after 24 and 72 h of treatment than **Hib-ester** in reducing RPMI 8226 cells growth, whereas (2R,3R/2S,3S)-**14** was 34-fold more active than **Hib-ester** after 24 h, although its efficacy decreased with time. During the design phase, structural simplification and synthetic accessibility issues were considered. The SAR evaluation identified 5-oxo-pyrrolidine scaffold as a potentially valuable framework for developing new agents against multiple myeloma cancer cells, and compounds with bulky substituents at C3 and a carboxylic acid, such as methyl ester, at C2 showed a good cytotoxicity. Worth noting is that 1(N),2,3 trisubstituted-5-oxopyrrolidines was easily obtained via the Castagnoli–Cushman reaction, considered as a green synthetic route thanks to the high atom economy of the reaction and the elimination of water as only a reaction side-product.

Lastly, we performed a preliminary investigation on the role of chirality in the activity against multiple myeloma. Although both chiral centers are essential for the cytotoxic activity, the biological investigation (MTT test and proteasome inhibition) of compound **13** in its enantiomeric forms, showed that the (*2R,3R*) enantiomer is the eutomer, but with a low eudysmic ratio. The obtained results suggest that in early-stage drug development, i.e., during the hit-to-lead process, racemic mixtures could be studied, thus providing a more economic and broad ecological advantage over single enantiomers, leading to the subsequent optimization stage and a deeper investigation of enantiomeric forms.

Further development of this class of compounds is ongoing.

## Data Availability

Not applicable.

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
