# Peer review of "From Nature to Synthetic Compounds: Novel 1(N),2,3 Trisubstituted-5-oxopyrrolidines Targeting Multiple Myeloma Cells"

_ijms, 2022, doi:10.3390/ijms232113061_

Round 1
Reviewer 1 Report
Listro et al. conducted synthetic work to yield a series of Hib-ester analogues, and subsequently these synthetic compounds were subjected to the antiproliferative bioassay. This work displayed a new chemotype as proteasome inhibitors, which could be interesting to readers. However, revisions are required if the article is considered for publication.
Comments:
1. Which compound exhibited better antiproliferative property, Hib-ester or 13 in your current study? As shown, all the synthetic compounds didn’t have a hydroxyl group but Hib-ester had. Maybe the existence of the hydroxyl group could improve the antiproliferative activity.
2. The synthetic procedure for N-methyl lactam esters (S/R)-7 includes the using of iodomethane in presence of metallic sodium. The iodomethane was highly toxic, and the metallic sodium was very dangerous to use. Did you try any other synthetic protocol?
3. For identification of the products, Rf values, mass data, elemental analysis data, and 13C NMR data are necessary. For the products containing heteroatom such as N, the IR data is required to confirm the existence of heteroatom bond such as C-N. Please add these data in the manuscript, and provide the MS, IR and NMR spectra as supplementary materials.
4. Why was only compound 13 separated by the chiral HPLC? It seemed insufficient to use only one pair of enantiomers to judge which enantiomer had better antiproliferative activity. Although the molecular docking experiment revealed the (2R,3R)-14 was more active than the opposite enantiomer, the screening experiments for both enantiomers of 14 would give more reliable results.
5. As shown in Figure 7, the interactions between compound 13 and the proteasome chymotrypsin were different from those for its methyl ester 14. Therefore, please check ‘Our modeling results elucidate that both compounds 13 and 14 are able to interact with the main chains of same pivotal residues of the proteasome chymotrypsin pocket...’. And please provide RMSD values for the molecular docking experiment.
6. What were positive controls in the bioassays?
Others:
1. Addition: P1L23: ‘Multiple Myeloma’ → ‘Multiple Myeloma (MM)’
2. Italics: P1L23: ‘Hibiscus Sabdariffa L.’ → ‘Hibiscus Sabdariffa L.’
3.Figure 1: There was no R4 in the chemical structure but an X. Please check it.
4. Scheme 1: ‘H2SO4’ → ‘H2SO4’ ‘OsO4, (0.2 equiv.)’ → ‘OsO4 (0.2 equiv.),’
5. To keep consistence, it is better to revise ‘eq’ as ‘eqiv.’ in the captions of S Schemes 2, 3 and 6.
6. P3L121: ‘S/R’ → ‘S/R’
7. Scheme 4: ‘CH3I’ → ‘CH3I’ ‘SOCl2’ → ‘SOCl2’
8. Scheme 5: ‘H2O/CH3OH’ → ‘H2O/CH3OH’
9. P5L191: ‘tr’ → ‘tR’
10. P15L514 & 519: ‘CH2’ → ‘CH2’
11. The pages of Ref. [9] was missing.
Please check typo errors throughout the whole manuscript.
Author Response
Reviewer 1
Listro et al. conducted synthetic work to yield a series of Hib-ester analogues, and subsequently these synthetic compounds were subjected to the antiproliferative bioassay. This work displayed a new chemotype as proteasome inhibitors, which could be interesting to readers. However, revisions are required if the article is considered for publication.
A: We thank the reviewer for the positive feedback and for the precious suggestions
Comments:
- Which compound exhibited better antiproliferative property, Hib-ester or 13in your current study? As shown, all the synthetic compounds didn’t have a hydroxyl group but Hib-ester had. Maybe the existence of the hydroxyl group could improve the antiproliferative activity.
A: As stated in the Conclusions, compound 13 resulted twelve-fold more active than Hib-ester in reducing RPMI 8226 cells growth. Moreover, compounds 10, 12, 14, 16, 18 and 20 showed IC50 values at least 6-fold lower than that of Hib-ester. These results support the hypothesis that the hydroxyl group is not fundamental for cytotoxic activity in the investigated cell lines. These aspects have been stressed in the reviewed manuscript.
- The synthetic procedure for N-methyl lactam esters (S/R)-7 includes the using of iodomethane in presence of metallic sodium. The iodomethane was highly toxic, and the metallic sodium was very dangerous to use. Did you try any other synthetic protocol?
A: The reviewer is right about the safety concerns of the protocol described. We tried other protocols, but they were unsuccessful. It has to be underlined that the applied procedures have been performed following standard safety practices, as usually performed in modern synthetic laboratories. The protocol employed is overall rather simple and gives the desired product in a more efficient way compared to other attempts experimented by us (e.g. use of Lithium diisopropylamide or NaH). Since our aim was to obtain the target molecule in a sufficient amount for the screening, we did not perform an in-depth investigation to optimize the synthetic protocol.
- For identification of the products, Rfvalues, mass data, elemental analysis data, and 13C NMR data are necessary. For the products containing heteroatom such as N, the IR data is required to confirm the existence of heteroatom bond such as C-N. Please add these data in the manuscript, and provide the MS, IR and NMR spectra as supplementary materials.
A: According to the Reviewer's suggestions, we have implemented the characterization of the final compounds, including Rf and MS values. Although the 13C-NMR spectra are not mandatory according to the Journal guidelines, we added the 13C-NMR of the most interesting compound (13 and 14).
- Why was only compound 13 separated by the chiral HPLC? It seemed insufficient to use only one pair of enantiomers to judge which enantiomer had better antiproliferative activity. Although the molecular docking experiment revealed the (2R,3R)-14 was more active than the opposite enantiomer, the screening experiments for both enantiomers of 14 would give more reliable results.
A: The Reviewer is right, but let us allow to explain our approach. Compound 13 resulted in the most promising scaffold (R=carboxyl group) and it will be further derivatized in a hit-to-lead program. With this regard, racemic 13 is not properly a scaffold, but a mixture of two scaffolds, i.e. the two enantiomers. Starting from this consideration, we performed the chiral separation and assigned the absolute configuration to enantiomeric 13. The biological investigation highlights that (2R/3R)-13 can be considered the enantiopure scaffold to be further investigated. In our ongoing research, it will be used for expanding the SAR.
- As shown in Figure 7, the interactions between compound 13and the proteasome chymotrypsin were different from those for its methyl ester 14. Therefore, please check ‘Our modeling results elucidate that both compounds 13 and 14 are able to interact with the main chains of same pivotal residues of the proteasome chymotrypsin pocket...’. And please provide RMSD values for the molecular docking experiment.
A: We modified the sentence as following “Our modeling results elucidate that both compounds 13 and 14 are able to interact with the main chains of some pivotal residues of the proteasome chymotrypsin pocket”
Regarding the RMSD we reported the value in Materials and Methods - Computational studies section.
- What were positive controls in the bioassays?
A: The positive control is Bortezomib. The respective description is now reported in page 9 line 273 and the experimental data is in the supplementary material (Figure S4)
Others:
- Addition: P1L23: ‘Multiple Myeloma’ → ‘Multiple Myeloma (MM)’
- Italics:P1L23: ‘Hibiscus Sabdariffa L.’ → ‘Hibiscus SabdariffaL.’
3.Figure 1: There was no R4 in the chemical structure but an X. Please check it.
- Scheme 1: ‘H2SO4’ → ‘H2SO4’ ‘OsO4, (0.2 equiv.)’ → ‘OsO4(0.2 equiv.),’
- To keep consistence, it is better to revise ‘eq’ as ‘eqiv.’ in the captions of SSchemes 2, 3and 6.
- P3L121: ‘S/R’ → ‘S/R’
- Scheme 4: ‘CH3I’ → ‘CH3I’ ‘SOCl2’ → ‘SOCl2’
- Scheme 5: ‘H2O/CH3OH’ → ‘H2O/CH3OH’
- P5L191: ‘tr’ → ‘tR’
- P15L514 & 519: ‘CH2’ → ‘CH2’
- The pages of Ref. [9] was missing.
Please check typo errors throughout the whole manuscript.
A: We thank the reviewer for pointing out these errors. The manuscript has been corrected accordingly.
Reviewer 2 Report
The manuscript by Listro et al. and colleagues was an interesting read and very well-structured. I recommend the following minor corrections:
1) The discussion section explaining the molecular interactions that the hit compounds make with proteasome chymotrypsin binding site should be explained in more detail and compared with reference ixazomib's interaction with the binding pocket.
2) Previous studies on other proteasome inhibitors should be cited and compared to make a stronger case of identified hits.
Overall, I recommend the manuscript for publication in IJMS after minor revisions.
Author Response
Reviewer 2
The manuscript by Listro et al. and colleagues was an interesting read and very well-structured. I recommend the following minor corrections:
1) The discussion section explaining the molecular interactions that the hit compounds make with proteasome chymotrypsin binding site should be explained in more detail and compared with reference ixazomib's interaction with the binding pocket.
A: We thank the reviewer, and we added this section in the manuscript:
By analyzing the binding mode of each enantiomer, we noticed that all compounds are involved in several interactions, such as hydrogen bond and salt bridges interactions with proteasome chymotrypsin active site (Figure 7) and all of them bind the pocket area in which also ixazomib binds.
In detail the (2R,3R)-13 establishes hydrogen bond interactions with Lys32 and Gln131 proteasome residues (Figure 7A). Moreover, (2R,3R)-13 is engaged in several hydrophobic contacts with Thr1, Arg19, Thr21, Lys32, Lys33, Ser129 and Tyr130. The (2S,3S)-13 is able to establish hydrogen bond interactions with the Thr1, Lys33 and Gly47 and a salt bridge with the Lys33 (Figure 7A). In addition, the (‒)-(2S,3S)-13 was found well stabilized in the chymotrypsin site by means of several hydrophobic contacts with Thr1, Arg19, Thr21, Lys33, Gly47, Ala 49, Ser129, Tyr130 and Gln131.
Regarding the (2R,3R)-14 it is engaged in two hydrogen bonds one with Ser129 and the second with Tyr130. Moreover, (2R,3R)-14 is engaged in several hydrophobic contacts with Arg19, Ala20, Lys33, Ser28, Met45, Ala49, Ser53, Ser129, Tyr130 and Gln131. Considering the enantiomer (2S,3S)-14 is able to establish a hydrogen bond with the Ala49, and a water bridge with Ala50 and additional several hydrophobic contacts with Thr1, Arg19, Ala20, Thr21, Ala27, Ala49, Ala50, Ala123, Ser129, Tyr130 and Gln131.
2) Previous studies on other proteasome inhibitors should be cited and compared to make a stronger case of identified hits.
A: During the biological investigation we evaluated Bortezomib as positive control. Thus we added the experimental description on page 9 line 273 and the results are reported in the supplementary material (Figure S4).
Overall, I recommend the manuscript for publication in IJMS after minor revisions
Round 2
Reviewer 1 Report
This manuscript has been improved clearly.
However, the definition of ‘X’ in Figure 1 was wrong, which was not consistent with that in Table 1.
And typo errors could be found. For examples, the second ‘tR1’ should be revised as ‘tR2’ (Page 5 Line 193). The symbols in ‘* p < 0.05 and ** p < 0.01’ were not properly displayed (Figure 5 Caption). Some ‘R’ and ‘S’ were in italics, some were not (Page 12).
Considering these errors could be corrected during the proof process. Consequently this manuscript could be accepted in the present form.